# RFC-GAN — Feedback-Conditioned GAN for Iterative Memory-to-Sketch Facial Translation

## Abstract

Eyewitness–to–sketch translation is traditionally performed by human artists, a process prone to bias and information loss. Recent work has applied conditional Generative Adversarial Networks (GANs) to automate this task, yet existing models remain limited in their ability to iteratively refine coarse recollections into photorealistic faces. We propose a context-feedback training paradigm for image-to-image GANs: at each step, the generator and discriminator receive the most recent output as an auxiliary three-channel input, enabling the model to reason over its own predictions. Building on the Pix2Pix framework, we further investigate where to embed Self-Attention and Edge-Gating modules within the encoder–decoder and skip connections, systematically analyzing their effect on perceptual and adversarial loss. Experiments on synthetic-sketch and real eyewitness datasets demonstrate consistent improvements in Fréchet Inception Distance (FID), Learned Perceptual Image Patch Similarity (LPIPS), and human-rated realism, with the model producing sharper, more faithful faces and exhibiting stable iterative refinement at inference. These results suggest that feedback-conditioned GANs provide a principled path toward reliable facial reconstruction from memory.

## 1 Introduction

Facial sketching based on eyewitness accounts remains a widely used tool in forensic investigations. Yet despite its prevalence, the sketch-to-identification pipeline is fraught with inaccuracy, subjectivity, and operational bottlenecks. Only 23–38% of sketches directly lead to arrests (Frowd, 2012), and over 60% of wrongful convictions involve eyewitness misidentification (Innocence Project, 2024). While iconic cases such as the Oklahoma City and Boston Marathon bombings have highlighted the potential utility of forensic sketches (Federal Bureau of Investigation, 2020; Bloomberg, 2013), these successes are rare exceptions in an otherwise brittle process.

The traditional eyewitness-to-sketch workflow is vulnerable at every stage: memory distortion under stress (Deffenbacher et al., 2004), subjective verbal articulation, and heavy reliance on the artist's interpretation and skill. Each step introduces uncertainty and amplifies prior noise. Redrawing is time-consuming, and studies show that recognition accuracy improves dramatically when sketches are replaced with photorealistic imagery (Adams, 2010). These factors collectively limit the forensic and legal reliability of current methods.

To address these shortcomings, we explore the potential of deep generative models to automate and enhance the sketch-to-photo translation process. While recent work has applied Conditional Generative Adversarial Networks (CGANs) to this task, existing models are fundamentally constrained by their one-shot generation paradigm. Once a photo is generated, there is no structured mechanism to revisit, refine, or improve upon the output. This sharply contrasts with how humans operate—iteratively revisiting drafts to resolve uncertainties.

In this paper, we introduce **RFC-GAN**, a feedback-conditioned generative framework designed to iteratively refine facial reconstructions from coarse sketches. Our model departs from feedforward CGANs by embedding Recursive Feedback Conditioning (RFC): at each training and inference step, the generator and discriminator receive the model's prior output as an auxiliary input. This feedback loop enables the network to reason over its own predictions—refining details over multiple passes and correcting earlier artifacts in context.

To further enhance fidelity and structure, we integrate two additional architectural modules:

- Self-Attention (SA) enables long-range spatial interactions within the U-Net generator and PatchGAN discriminator, promoting consistency across facial features.
- Edge-Gating (EG) leverages sketch-derived edge maps to modulate encoder and skip-pathway activations, improving contour fidelity and boundary sharpness.

The remainder of this paper is organized as follows: Section 2 reviews related work in conditional generation and sketch translation. Section 3 details the RFC-GAN architecture, loss formulation, and the RFC, SA, and EG modules. Section 4 expands on the implementation of each component. Section 5 and Section 6 describe our datasets, training setup, and evaluation metrics, respectively. Section 7 presents quantitative and qualitative results, followed by conclusions and future directions in Section 8.

## 2 RELATED WORK

Deep generative models have advanced rapidly in recent years. Isola et al. (2017) introduced Pix2Pix, a supervised model requiring paired data, but its baseline version suffers from unstable convergence and mode collapse, where outputs become overly similar across inputs (Brownlee, 2019). To address data requirements, Zhu et al. (2017) proposed CycleGAN, an unsupervised framework using cycle-consistency to map between domains without paired datasets, though such models often face training instability (Hoyez et al., 2022).

Several extensions have sought to improve realism and flexibility. Discriminative Region Proposal Adversarial Networks (DRPAN) (Wang et al., 2018) enhanced local realism with reviser and region-proposal networks but added complexity, limiting adaptability. SelectionGAN (Tang et al., 2019) employed multi-channel attention for cross-view translation but struggled with fine-grained facial details. Multimodal UNsupervised Image-to-image Translation (MUNIT) (Lin et al., 2019) offered multimodal outputs at the cost of photorealism, while StarGAN (Choi et al., 2018) enabled multi-domain translation but required careful tuning to preserve identity features. EdgeGAN (Tang et al., 2020) leveraged edge maps to improve detail, and Laplacian Pyramid Generative Adversarial Network (LPGAN) (Denton et al., 2015) pioneered progressive refinement through multi-stage generation, stabilizing convergence and improving fidelity.

Building on these advances, we propose a GAN architecture tailored for sketch-to-image facial translation. Our model integrates a dynamic training schedule to mitigate mode collapse through staged activation of components (e.g., RFC, EG). The central innovation is RFC, which feeds previous outputs back into both generator and discriminator, enabling iterative refinement during training and inference. Complementary modules such as self-attention and edge-gating were also tested individually and with RFC, consistently showing that RFC delivers the most significant performance gains.

## 3 METHODOLOGY

This section describes our training setup, including the composite loss function and architectural enhancements. We also adopt three key components: RFC, SA, and EG, targeting spatial coherence, global context, and boundary precision.

### 3.1 LOSS FORMULATION AND TRAINING OBJECTIVE

To train our generator, we combine adversarial, reconstruction ($\ell_1$), and perceptual losses, balancing them to encourage accurate photo-reconstruction, pixel-level fidelity, and realism.

Given $\hat{y} = G_\theta(s)$, we adopt patchwise non-saturating losses as follows:

$$\mathcal{L}_D = \tfrac{1}{2} \operatorname{BCE}\big(D_\phi(s,\hat{y}), 0\big) + \tfrac{1}{2} \operatorname{BCE}\big(D_\phi(s,y), 1\big), \tag{1}$$

$$\mathcal{L}_G = \lambda_{\mathrm{adv}} \operatorname{BCE}\big(D_\phi(s,\hat{y}), 1\big) + \lambda_l \left\| \hat{y} - y \right\|_1 + \lambda_{\mathrm{perc}} \left\| \varphi(\hat{y}) - \varphi(y) \right\|_2^2, \tag{2}$$

where: $G_\theta$ is the generator with parameters $\theta$, taking a structured input $s$ and producing an output $\hat{y}$, $D_\phi$ is the discriminator with parameters $\phi$, which scores the realism of a pair $(s, y)$ or $(s, \hat{y})$, $\mathrm{BCE}(\cdot, \cdot)$ denotes the binary cross-entropy loss, $\|\hat{y} - y\|_1$ is the pixel-wise $\ell_1$ reconstruction loss, $\varphi(\cdot)$ represents features extracted from a pretrained VGG-19 network, and $\lambda_{\mathrm{adv}}, \lambda_l, \lambda_{\mathrm{perc}}$ are weighting factors for adversarial, reconstruction, and perceptual losses, respectively.

The discriminator is trained to distinguish between real and generated samples using the standard binary cross-entropy formulation. The generator, in turn, is guided by an adversarial term that aims to fool the discriminator into classifying its outputs as real. The reconstruction loss, $\|\hat{y} - y\|_1$, enforces pixel-level alignment between generated and target images. Increasing $\lambda_l$ strengthens pixel supervision but may lead to overly smooth or blurred outputs, as the generator learns to average pixel intensities. On the other hand, reducing $\lambda_l$ allows for sharper textures at the cost of potential structural artifacts—particularly in semantically dense regions like facial features. The perceptual loss, $\|\varphi(\hat{y}) - \varphi(y)\|_2^2$, is computed over intermediate features of a pretrained VGG-19 network and encourages high-level semantic similarity beyond raw color correspondence. We empirically set $\lambda_{\mathrm{adv}}{=}3$, $\lambda_l{=}50$, $\lambda_{\mathrm{perc}}{=}0.3$ to ensure balanced gradients across the loss terms and avoid training collapse due to dominance of any single term.

## 3.2 ARCHITECTURAL COMPONENTS

To investigate their effectiveness individually and in combination, we adopt three architectural features for our model:

### 3.2.1 RFC

RFC is a self-conditioning mechanism that recursively feeds the model's previous output back into itself. It enables progressive refinement by maintaining contextual awareness over iterations. During training, RFC allows the generator to revisit and improve upon earlier predictions, similar to how an artist references prior sketches when adding detail. Let $s_i$ denote the input sketch for sample $i$. At training step $t$, the output is defined recursively as:

$$\hat{y}_i^{(t)} = r_i^{(t)} = G_\theta\big(s_i \oplus r_i^{(t-1)}\big), \qquad r_i^{(0)} = \mathbf{0}, \tag{3}$$

where $\oplus$ denotes channel-wise concatenation. The discriminator $D_\phi$ receives the tuple $(\hat{y}_i^{(t)}, s_i, r_i^{(t-1)})$.

To avoid backpropagation through time, we maintain a per-sample buffer of $r_i^{(t-1)}$ updated with `stop-gradient`, ensuring the previous outputs do not receive gradients. The per-step training objectives for the generator and discriminator are:

$$\mathcal{L}_G^{(t)} = \lambda_{\mathrm{adv}} \, \mathbb{E}\Big[-\log D_\phi\big(\hat{y}_i^{(t)}, s_i, r_i^{(t-1)}\big)\Big] + \lambda_1 \big\|\hat{y}_i^{(t)} - y_i\big\|_1 + \lambda_{\mathrm{perc}} \big\|\varphi(\hat{y}_i^{(t)}) - \varphi(y_i)\big\|_2^2, \tag{4}$$

$$\mathcal{L}_D = \tfrac{1}{2}\,\mathrm{BCE}\big(D_\phi(\hat{y}_i^{(t)}, s_i, r_i^{(t-1)}), 0\big) + \tfrac{1}{2}\,\mathrm{BCE}\big(D_\phi(y_i, s_i, r_i^{(t-1)}), 1\big). \tag{5}$$

RFC naturally induces a fixed point $\hat{y}^* = G_\theta(s \oplus \hat{y}^*)$. At inference time, we iteratively update:

$$\hat{y}^{(t)} \leftarrow G_\theta(s \oplus \hat{y}^{(t-1)}),$$

terminating either after a fixed number of steps $T$ or when convergence is reached. The optimal number of refinement steps is selected empirically (see Section 7).

### 3.2.2 SA

SA is a module that captures long-range spatial dependencies by allowing each pixel to attend to all others. It is integrated into both the U-Net generator and PatchGAN discriminator to promote global coherence, especially across distant regions of the image.

### 3.2.3 EG

EG is a gating mechanism designed to enhance edge and boundary fidelity. It is applied to either encoder blocks or skip connections within the U-Net and helps sharpen contours by selectively emphasizing edge information during generation.

### 3.3 EVALUATION METRICS

We evaluate each module independently and in combination. For RFC, we assess the effect of introducing feedback from the start of training versus delayed integration. For SA, we examine different placement strategies within the U-Net hierarchy to balance expressiveness and regularization. For EG, we compare gating configurations along encoders, skip connections, or both.

To formalize our experimental goals, we define the following hypotheses:

- **H1 (RFC Timing):** *The model achieves stronger early validation performance and improved final image quality when RFC is applied from the first epoch of training*, leveraging iterative self-correction throughout the learning process.

- **H2 (Minimal-SA):** The model yields higher perceptual fidelity when a minimal number of Self-Attention (SA) blocks are introduced at targeted locations, as opposed to deeper or denser attention stacks, which risk over-smoothing and redundancy.

- **H3 (EG Effect):** The model exhibits improved boundary accuracy and sharper contour localization when EG is incorporated into the encoder or skip connections, without sacrificing global image consistency.

- **H4 (Complementarity):** The model achieves the highest overall performance when RFC, SA, and EG are used in combination, confirming their complementary benefits in refining structure, context, and edge detail.

## 4 MODEL ARCHITECTURE

A key limitation of vanilla Pix2Pix (Isola et al., 2017) in sketch-to-photo translation lies in its training instability (e.g., mode collapse) and its inability to model long-range spatial dependencies. These issues often lead to unrealistic outputs and poor identity preservation. To address these challenges, we introduce three architectural components designed to enhance generation quality and improve model stability: RFC, SA, and EG.

### 4.1 RECURSIVE FEEDBACK CONDITIONING

RFC augments the generator's input by feeding back the previous output as an additional 3-channel tensor. To implement RFC, we preallocate three channels in the input to both $G$ and $D$, initialized to zero. The buffer stores full-resolution RGB outputs, indexed by sample ID and refreshed per iteration or epoch. We consider two RFC activation strategies: (1) starting from the beginning of training (epoch 0), and (2) delaying RFC activation to mid-training. Intuitively, enabling RFC from epoch 0 will lead to better performance than it's mid-start counterpart, as delaying its introduction may train the generator to treat the three-channel feedback tensor as an always-zero placeholder prior to activation, causing it to ignore the signal even after it becomes informative.

### 4.2 SELF-ATTENTION

To inject global context, we evaluate SA placement strategies mainly within the U-Net generator. For our simulations, we attempted three general variations of SA placements. Each variation had the option of including self-attention in the bottleneck, in addition to the option of mirroring attention on the decoder blocks - and thus making the entire model symmetrically attentive. As for the discriminator, we opted to retain the SA module in the lowest possible layer, as design choices are limited and our primary focus lies in exploring the generator's architecture. Figure 1 summarizes the full ablation tree. Panel (b) enumerates the three SA scheduling strategies - consecutive (option 1), alternating (option 2), and minimal (option 3) - while panel (a) expands each strategy into two architectural variants (with self-attention bottleneck vs. without), and each of those into two symmetry choices (attention-symmetric vs. asymmetric), yielding the complete set of configurations analyzed. For example, `SA12` denotes the *minimal* attention choice: no self-attention at the bottleneck and an *asymmetric* layout, with attention applied only at the decoder's sixth block (d6).

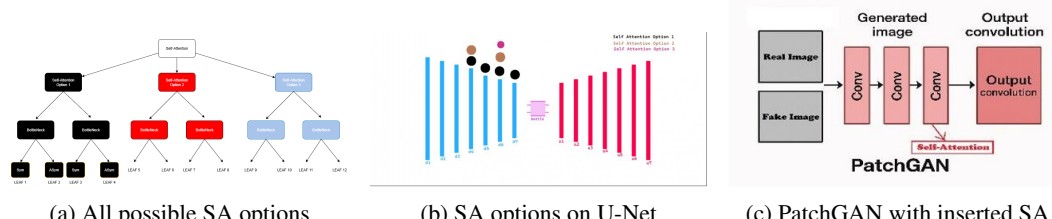

(a) All possible SA options  (b) SA options on U-Net  (c) PatchGAN with inserted SA

Figure 1: **Self-Attention simulations.** Each leaf in the placement tree corresponds to a specific insertion point, labeled SA1-SA12. For example, SA10 refers to the tenth leaf node in the tree, which matches a particular bottleneck/skip location with a certain SA option (as shown in (b)).

### 4.3 EDGE-GATING

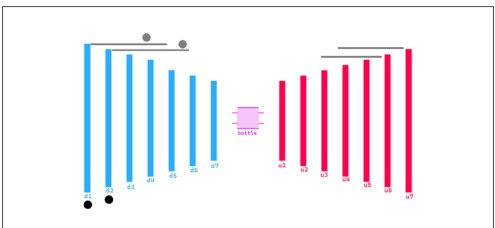

Figure 2: **Edge-Gating (EG) configurations.** EG1 applies gating at the encoder's first two layers ($d1$, $d2$). EG2 applies gating at the first two skip connections (skip1, skip2). EG3 combines both strategies, gating $d1$–$d2$ as well as skip1–skip2.

We evaluate three edge-gating (EG) placements using a sketch's edge map $E(s)$: **EG1** injects edges into the discriminator's first two blocks (d1,d2), **EG2** injects into the generator's early skip connections (skip1,skip2), and **EG3** applies both, coupling edge awareness in generator and discriminator. We introduce EG at epoch 10 of a 20-epoch run, the experimental cap chosen to standardize our benchmarks. The delayed start is intended as the model's warm-up, to first accustom it to learning the coarse sketch-to-structure mapping, so that when EG is enabled it acts as a boundary-refinement prior rather than the primary driver of the generator's outputs

## 5 DATASETS

The first dataset used in our study was the Person Face Sketches dataset from Kaggle (Lee et al., 2020), which contains over 21,000 facial images from CelebAMask-HQ, each paired with a synthetic sketch. While the dataset offers consistent and well-aligned (sketch, photo) pairs suitable for supervised training, its synthetic nature - mainly due to the sketches being artificially generated - introduces a domain constraint. Specifically, the generated sketches lack the variability, imprecision, and stylistic noise typically found in hand-drawn inputs—such. These characteristics limit the dataset's representativeness for downstream real-world use cases, particularly in forensic scenarios involving human-drawn sketches.To address the domain discrepancy, we standardized all sketches using a preprocessing filter based on edge detection techniques (Canny, 1986; Xie & Tu, 2015). This filter was applied uniformly to both training sketches and hand-drawn inputs at inference time, ensuring the model always receives sketches in a consistent and normalized format—grayscale, uniform stroke width, and fixed resolution. For benchmark testing, we employed the CUHK Face Sketch dataset (Wang & Tang, 2009; Zhang et al., 2011), which consists of 370 hand-drawn sketch-photo pairs created by artists. We evaluated the benchmark data with its own benchmarks: FID, Kernel Inception Distance (KID), LPIPS, Structural Similarity Index Measure (SSIM), Peak Signal-to-Noise Ratio (PSNR), and Sliced Wasserstein Distance (SWD) (Heusel et al., 2017; Zhang et al., 2018; Simonyan & Zisserman, 2015; Johnson et al., 2016)—across both datasets to enable consistent performance comparison. Samples of the datasets and augmentation methods are shown in Figure 6

# 6 EXPERIMENTAL SETUP AND EVALUATION METRICS

## 6.1 TRAINING SETUP

We train our models using mini-batches of size 8 and the Adam optimizer with a learning rate of $2 \times 10^{-4}$ for both generator and discriminator. The RFC and EG components are activated through a training scheduler, which controls an alpha parameter for each component, switching its contribution off (multiplying the variable channel by 0s) or on (multiplying the variable channel by 1s) depending on the training phase. A detailed description of the full training loop and RFC-update procedure is provided in Appendix B at Algorithm 1 .

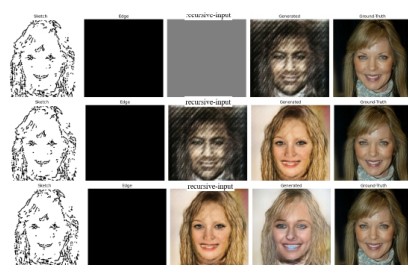
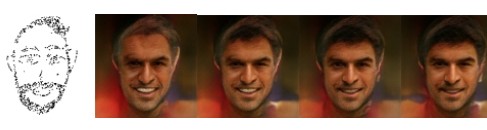

(a) Recursive input update during training.
(b) Recursive conditioning applied at inference.

Figure 3: Illustrations of recursive feedback conditioning (RFC). (a) shows how the recursive input is updated during training, while (b) demonstrates its effect at inference time.

## 6.2 METRICS

We evaluate generated images using complementary quantitative metrics that measure distributional alignment, perceptual similarity, and pixel-level fidelity. The following evaluation metrics were adopted:

- **FID (Fréchet Inception Distance):** Measures the distributional similarity between generated and real images in a deep feature space. Lower FID indicates closer alignment to the target data distribution.

- **KID (Kernel Inception Distance):** Similar to FID but uses a polynomial kernel over Inception features, and has the advantage of being unbiased even with small sample sizes.

- **LPIPS (Learned Perceptual Image Patch Similarity):** Evaluates perceptual similarity by comparing deep features of real and generated images, providing a proxy for human judgment of visual quality.

- **SWD (Sliced Wasserstein Distance):** Captures distributional differences across image patches through random projections, emphasizing multi-scale texture consistency.

- **SSIM (Structural Similarity Index):** Quantifies structural similarity between images, focusing on luminance, contrast, and texture; higher SSIM indicates closer structural alignment.

- **PSNR (Peak Signal-to-Noise Ratio):** A pixel-level fidelity metric that measures reconstruction quality relative to the ground truth; higher PSNR corresponds to lower distortion.

# 7 RESULTS AND ANALYSIS

For our experimental procedure, we followed two complementary dimensions, with all simulations being tested against the vanilla Pix2Pix model. The first evaluation front is a validation-loss analysis while the second is a quantitative benchmarking analysis on standard metrics (FID, KID, LPIPS, SSIM, PSNR , SWD). It is important to reiterate that all benchmarking was done on a separate evaluation dataset, distinct from the training set.

## 7.1 RECURSIVE FEEDBACK CONDITIONING

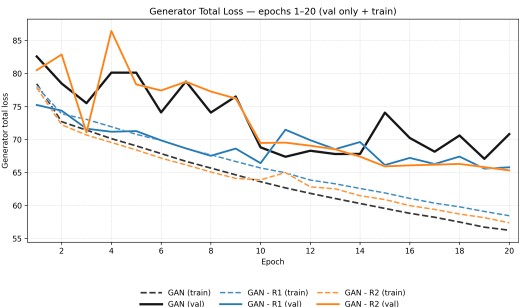

Figure 4: Training losses for R1 (RFC introduced at epoch 0) and R2 (RFC introduced at epoch 10), showing faster convergence when recursion is enabled from the start.

As shown in Figure 4, we can see both runs R1 and R2, outperform the baseline simulation. In fact, R1-like simulations, simulations that begin RFC at epoch 0) always begin at a significantly lower validation loss.

| Model | Steps | FID↓ | KID↓ | LPIPS↓ | SWD↓ | SSIM↑ | PSNR↑ |
|---|---|---|---|---|---|---|---|
| Pix2Pix | 0 | 282.35 | $0.383 \pm 0.005$ | $0.381 \pm 0.052$ | $0.190 \pm 0.134$ | $0.277 \pm 0.052$ | $8.46 \pm 1.18$ |
| RFC-GAN_R1 | 0 | 288.59 | $0.385 \pm 0.004$ | $0.329 \pm 0.049$ | $0.160 \pm 0.128$ | $0.359 \pm 0.063$ | $9.76 \pm 1.27$ |
| RFC-GAN_R1 | 5 | 214.33 | $0.260 \pm 0.003$ | $0.352 \pm 0.068$ | $0.179 \pm 0.141$ | $0.413 \pm 0.069$ | $9.22 \pm 1.63$ |
| RFC-GAN_R1 | 10 | 197.57 | $0.238 \pm 0.003$ | $0.361 \pm 0.074$ | $0.210 \pm 0.164$ | $0.425 \pm 0.065$ | $9.03 \pm 1.74$ |
| RFC-GAN_R1 | 15 | 198.88 | $0.246 \pm 0.004$ | $0.371 \pm 0.072$ | $0.236 \pm 0.178$ | $0.428 \pm 0.055$ | $8.79 \pm 1.70$ |
| RFC-GAN_R1 | 20 | 212.86 | $0.279 \pm 0.004$ | $0.384 \pm 0.067$ | $0.262 \pm 0.188$ | $0.424 \pm 0.045$ | $8.45 \pm 1.58$ |

Table 1: Results comparing the baseline GAN with recursive conditioning (R1) at different steps, showing significant improvement.

The results from Figure 4 and Table 1 show that introducing RFC at the first epoch significantly improves the model's stability and overall convergence. We can see that the best version of the R1 model is at step 10 - which means the stage at which we recursively fed the generator its previous output 10 times. The FID, KID, LPIPS, SSIM, and PSNR benchmarks reflect RFC's positive effects. However, the SWD metric, slightly dips. We believe this arises from the model prioritizing global fidelity and identity preservation. Regardless, the overall results discussed here, and in other the ensemble simulations that include an R1 RFC component, provide strong evidence in support of hypothesis *H1*.

## 7.2 SELF-ATTENTION (SA)

Before establishing our epoch phases, it is important to determine the layer-locations of including self-attention in our generator and discriminator. We found it intuitive to introduce self-attention modules in UNET's down sampling modules, as to capture and generate long-range dependency/symmetry across the image. So, it is a step further convolutions' special locality that is limited to filters and strides. As self-attention is expensive to have in the earlier layers, we opted to sparse them out between the middle and lower layers of the down sampling section.

The first variation, as shown in part (a) of Figure 8, was set to test if consecutive self-attention would improve performance. The second variation, also Figure 8, was an attempt to alternate self-attention modules throughout the U-Net architecture. The third and final variation applied SA minimally, distributing the modules at selected depths of the U-Net architecture. From a validation-curve inspection, placing self-attention layers consecutively, as Figure 8 shows, always led to underperforming models. Though better than the prior, the second simulation, Figure 8's second image, also underperformed, when compared to the baseline. We can see a change in trend in the final simulation in Figure 8, the simulation where we sparsely inserted self-attention. It narrowly beat the baseline generator by the end of the 20th epoch.

The results from Table 2 show that, although the validation loss curves suggested SA9, SA10, and SA12 to be feasible models, the benchmark metrics tell a different story. The FID, KID, LPIPS,

| Model | Steps | FID↓ | KID↓ | LPIPS↓ | SWD↓ | SSIM↑ | PSNR↑ |
|---|---|---|---|---|---|---|---|
| Pix2Pix | 0 | 282.35 | $0.383 \pm 0.005$ | $0.381 \pm 0.052$ | $0.190 \pm 0.134$ | $0.277 \pm 0.052$ | $8.46 \pm 1.18$ |
| RFC-GAN_SA6 | 0 | 301.20 | $0.410 \pm 0.005$ | $0.371 \pm 0.051$ | $0.179 \pm 0.127$ | $0.340 \pm 0.062$ | $8.70 \pm 1.21$ |
| RFC-GAN_SA7 | 0 | 305.70 | $0.422 \pm 0.005$ | $0.381 \pm 0.052$ | $0.179 \pm 0.132$ | $0.252 \pm 0.040$ | $8.47 \pm 1.20$ |
| RFC-GAN_SA9 | 0 | 310.00 | $0.418 \pm 0.005$ | $0.394 \pm 0.052$ | $0.181 \pm 0.124$ | $0.335 \pm 0.073$ | $8.16 \pm 1.16$ |
| RFC-GAN_SA10 | 0 | 325.37 | $0.447 \pm 0.006$ | $0.411 \pm 0.063$ | $0.210 \pm 0.167$ | $0.285 \pm 0.070$ | $7.82 \pm 1.35$ |
| RFC-GAN_SA12 | 0 | 323.29 | $0.436 \pm 0.008$ | $0.393 \pm 0.064$ | $0.183 \pm 0.135$ | $0.300 \pm 0.069$ | $8.22 \pm 1.44$ |

Table 2: Benchmark results for self-attention variants. Although SA6, SA7, SA9, SA10, and SA12 appeared most promising based on validation curves, all of these self-attention-only models ultimately failed to surpass the baseline in the benchmark tests.

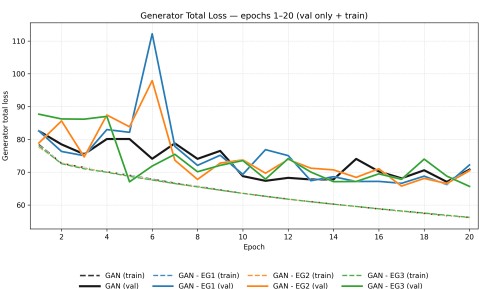

Figure 5: Edge-gating options validation loss curves.

SSIM, and PSNR values show that self-attention-only models greatly under perform, most notably SA10. It is important to note, however, that certain variants (e.g. SA9 and SA12) may still hold promise if combined with complementary techniques, such as RFC and EG. This does not mean a complete rejection of Hypothesis *H2* just yet; it is essential to analyze SA's role within component ensembles.

## 7.3 EDGE GATING (EG)

This edge that we have extracted is then used as a gating mask that helps the flow of features - specifically the sketch edges - within the U-Net generator, a method introduced in the paper by Tang et al. (2020). Upon activation; the generated images have more defined edges.

According to the validation loss curves, we can see that in Figure 7, the three edge gating options outperform the vanilla generator.

| Model | Steps | FID↓ | KID↓ | LPIPS↓ | SWD↓ | SSIM↑ | PSNR↑ |
|---|---|---|---|---|---|---|---|
| Pix2Pix | 0 | 282.35 | $0.383 \pm 0.005$ | $0.381 \pm 0.052$ | $0.190 \pm 0.134$ | $0.277 \pm 0.052$ | $8.46 \pm 1.18$ |
| RFC-GAN_EG1 | 0 | 331.11 | $0.465 \pm 0.005$ | $0.385 \pm 0.059$ | $0.231 \pm 0.165$ | $0.275 \pm 0.037$ | $8.39 \pm 1.37$ |
| RFC-GAN_EG2 | 0 | 321.29 | $0.445 \pm 0.006$ | $0.392 \pm 0.064$ | $0.250 \pm 0.177$ | $0.282 \pm 0.039$ | $8.26 \pm 1.48$ |
| RFC-GAN_EG3 | 0 | 248.94 | $0.326 \pm 0.003$ | $0.356 \pm 0.051$ | $0.183 \pm 0.143$ | $0.357 \pm 0.058$ | $9.07 \pm 1.24$ |

Table 3: Edge-gating benchmark results compared to baseline GAN.

In Table 3, there is one outlier simulation that outperforms the typical GAN generator: EG3. Gating the edge from the encoder layers and the skip connections together is the option that results in less blurry, more outlined images. In these tests, we also supplied the edge map to the deployed GAN, to faithfully recreate its training conditions. Hypothesis *H3* is corroborated by the validation-loss curves of EG3 and its benchmark tests.

## 7.4 COMPONENT ENSEMBLES

Hypothesis *H4* is intuitive, suggesting that a combination of the best version of our components would give us a model with better overall results than any of its individual components that produced the best simulations of their categories. As such, we proposed combining self-attention's SA7, SA9, and SA12 with edge-gating's EG3 and RFC's R1. Compared to the baseline GAN, combining SA and EG alone produced mixed results, performing worse than the baseline when more self-attention modules (SA7) were included, but outperforming the baseline when self-attention was inserted more sparsely (SA12). As such, SA12-EG3 combination led to strong results, but we hypothesize that

incorporating an RFC component to the SA-EG mix would further improve the model. We find this especially true, as the R1 led to the absolute best results in the benchmark tests-most clearly observed when comparing GAN-R1 in Figure 4 and Table 1 against all other samples presented as shown in Figures 6 and 7 and Tables 2, 3, and 5. Benchmark results and validation loss curves are provided in Appendix C (see Table 5 and Figure 8).

When observing the simulations that included the RFC's R1 component, we can clearly see that all generators improved significantly between the 5th and 15th epochs (excluding the SWD metric). This further reinforces Hypothesis *H1*, which highlights the strength of recursively feeding the model's previous output as a form of contextual information. The best simulation between the displayed 18 simulations was the SA12-EG3-R1 combinations, exactly as we anticipated. At the 5th step of our RFC process, the model peaks. This three-way ensemble of components outperforms its pairwise counterparts, supporting Hypothesis *H4*. However, the fact that the full ensemble does not surpass the R1-only simulation weakens the hypothesis' support, even if the differences in benchmark metrics are relatively narrow.

## 7.5 RESULTS

| Model | Steps | FID↓ | KID↓ | LPIPS↓ | SWD↓ | SSIM↑ | PSNR↑ |
|---|---|---|---|---|---|---|---|
| Pix2Pix | 0 | 282.35 | $0.383 \pm 0.005$ | $0.381 \pm 0.052$ | $0.190 \pm 0.134$ | $0.277 \pm 0.052$ | $8.46 \pm 1.18$ |
| RFC-GAN_R1 | 10 | 197.57 | $0.238 \pm 0.003$ | $0.361 \pm 0.074$ | $0.210 \pm 0.164$ | $0.425 \pm 0.065$ | $9.03 \pm 1.74$ |
| RFC-GAN_R1 | 15 | 198.88 | $0.246 \pm 0.004$ | $0.371 \pm 0.072$ | $0.236 \pm 0.178$ | $0.428 \pm 0.055$ | $8.79 \pm 1.70$ |
| RFC-GAN_SA12_EG3_R1 | 5 | 218.60 | $0.277 \pm 0.003$ | $0.370 \pm 0.076$ | $0.222 \pm 0.168$ | $0.366 \pm 0.070$ | $8.82 \pm 1.76$ |

Table 4: Benchmark results comparing the baseline GAN to top model contenders.

The best two models from the benchmark tests are the following: R1 and SA10-EG3-R1, and as shown on Table 5, the model with the best benchmarks - as well as the best validation-loss curve - is the RFC-only R1 model. The variant SA12–EG3–R1 ensemble demonstrates somewhat weaker performance. However, there are some important notes regarding this three-component model. It shows that sparsely placed self-attention, something also seen in SA12-EG3, is better than its other counterparts IF COMBINED WITH EG3. This somewhat supports hypothesis *H2*, however, we cannot fully corroborate it, as SA12 alone does not give better results that the baseline GAN. Hypothesis *H2* is supported from Figure 7 and Table 3.Hypothesis *H3* is supported as edge-gating (when applied singularly) improves image fidelity, when compared to the baseline measure. Hypothesis *H4* isn't fully validated, as the as SA12–EG3–R1 trails the R1-only model in overall performance. Nevertheless, it remains the second-strongest model, indicating that while the full ensemble does not yield the absolute best performance, it still provides substantial support for the underlying intuition of hypothesis *H4*.

## 8 CONCLUSION AND FUTURE WORK

Recursive Feedback Conditioning consistently improved image-reconstruction quality, during training as well as at inference time. Whenever the R1 component—i.e., the RFC variant where feedback conditioning is introduced at the first epoch — were observed to give significantly stronger benchmark results: this validates our original H1 Hypothesis. Edge-gating, especially EG3—which gates $d_1$, $d_2$, and the first two skip connections—proved most effective in sharpening edges and reducing blur, thereby strongly supporting Hypothesis H3. While self-attention on its own consistently underperformed, certain sparsely placed SA layers, when used in ensembles with RFC or EG, did contribute to stronger results. This means that H2 partially fails: self-attention alone does not improve over the baseline, but in some ensemble-combinations it outperforms the benchmark. The benchmark analysis highlights R1 as the strongest overall model, with SA12–EG3–R1 ensemble being the second best. One possible explanation for the ensemble's "failure" is that the combination of components introduces a bottleneck effect in the model's network, with multiple components working together may restrict information flow rather than enhance it - or even propagate redundant information.

For future work, we note that the RFC-only model only has a limited, short-term memory. This leaves us with the opportunity of exploring deeper recursive structures, structures that introduce multiple previously outputted images to both the generator and discriminator. By doing so, the model would gain longer contextual memory, thus possibly giving us better results.

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

## A  ADDITIONAL DETAILS AND RESULTS

### A.1  ETHICS STATEMENT

This system produces photorealistic faces from sketches and is not validated for identification or forensic decision-making; any high-stakes use requires task-specific evaluation, IRB/ethics approval, and human oversight. We restrict training and evaluation to consented data, apply data minimization and retention limits, and avoid scraped faces without permission. To mitigate demographic harms, we recommend reporting subgroup metrics when feasible and auditing for domain shift (synthetic vs. hand-drawn sketches); nonetheless, residual disparities may persist. To reduce misuse risks (e.g., impersonation, defamation), we advocate a research-only license, visible/invisible watermarking, audit logs (including seeds and iteration counts), and optional out-of-distribution rejection with confidence cues. We will document model scope and failure modes via model/datasheet cards and deprecate checkpoints that exhibit unacceptable subgroup harm. Finally, we disclose compute and encourage reproduction with fixed seeds and versioned environments to promote transparency and minimize environmental impact.

### A.2  RESULTS

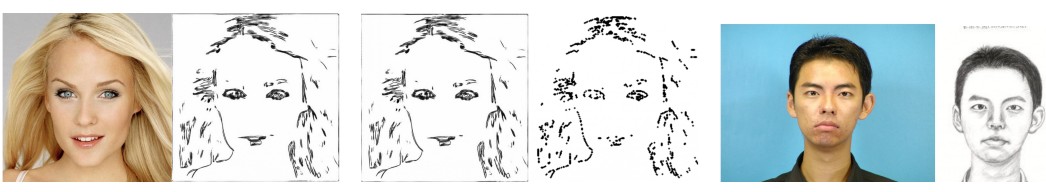

(a) Unfiltered Training Dataset Pair  (b) Pre/PostProcess Pair  (c) Benchmark Dataset Pair

Figure 6: Sample of different pairwise data in our two datasets, as well a sample of how our preprocessing filter outputs.

---

**Algorithm 1** Training with RFC and Edge-Gating (abstract)

---

1: **Hyperparameters:** StartEdgeGatingEpoch, StartRFCEpoch, $N$
2: **Channels:** $C_{\text{sketch}}=3$, $C_{\text{edge}}=1$, $C_{\text{rec}}=3$, $C_{\text{true}}=3$
3: $C_G \leftarrow C_{\text{sketch}} + C_{\text{edge}} + C_{\text{rec}}$;    $C_D \leftarrow C_{\text{sketch}} + C_{\text{edge}} + C_{\text{rec}} + C_{\text{true}}$
4: **Optimization:** mini-batch size $B=8$;    $\text{Opt}_G \leftarrow \text{ADAM}(2 \times 10^{-4})$;  $\text{Opt}_D \leftarrow \text{ADAM}(2 \times 10^{-4})$
5: ZeroImage3ch $\leftarrow$ ZEROSLIKE(3 channels)
6: **for all** sketch in dataset **do**
7:     PreviousOutputMap[sketch_id] $\leftarrow$ ZeroImage3ch
8:     EdgeMapCache[sketch_id] $\leftarrow$ COMPUTEEDGE(sketch)
9: edgeAlpha $\leftarrow$ 0;   rfcAlpha $\leftarrow$ 0
10: **for** epoch $\leftarrow$ 0 **to** $N-1$ **do**
11:     **if** epoch $\geq$ StartEdgeGatingEpoch **then**
12:         edgeAlpha $\leftarrow$ 1
13:     **if** epoch $\geq$ StartRFCEpoch **then**
14:         rfcAlpha $\leftarrow$ 1
15:     GenOutputBuffer $\leftarrow$ $\varnothing$
16:     **for all** (sketch_id, ( sketch_rgb, edge_1ch, true_rgb )) $\in$ BATCH($B$) **do**
17:         RecIn $\leftarrow$ rfcAlpha $\cdot$ PreviousOutputMap[sketch_id]
18:         EdgeMap $\leftarrow$ edgeAlpha $\cdot$ EdgeMapCache[sketch_id]
19:         GenInput $\leftarrow$ CONCAT(sketch_rgb, EdgeMap, RecIn)
20:         GenOutput $\leftarrow$ GENERATOR(GenInput)
21:         GenOutputBuffer[sketch_id] $\leftarrow$ GenOutput
22:     **for all** (sketch_id, ( sketch_rgb, edge_1ch, true_rgb )) $\in$ BATCH($B$) **do**
23:         RecIn $\leftarrow$ PreviousOutputMap[sketch_id]
24:         FakeImg $\leftarrow$ GenOutputBuffer[sketch_id]
25:         DiscInput_Real $\leftarrow$ CONCAT(sketch_rgb, edgeAlpha $\cdot$ edge_1ch, RecIn, true_rgb)
26:         DiscInput_Fake $\leftarrow$ CONCAT(sketch_rgb, edgeAlpha $\cdot$ edge_1ch, RecIn, FakeImg)
27:         DiscOut_Real $\leftarrow$ DISCRIMINATOR(DiscInput_Real)
28:         DiscOut_Fake $\leftarrow$ DISCRIMINATOR(DiscInput_Fake)
29:     EXCHANGELOSSESANDTRAIN($\text{Opt}_G$, $\text{Opt}_D$, $B$)
30:     **for all** sketch_id $\in$ GenOutputBuffer **do**
31:         PreviousOutputMap[sketch_id] $\leftarrow$ GenOutputBuffer[sketch_id]

---

# B  TRAINING SETUP PSEUDOCODE

# C  EXTENDED ABLATIONS

| Model | Steps | FID↓ | KID↓ | LPIPS↓ | SWD↓ | SSIM↑ | PSNR↑ |
|---|---|---|---|---|---|---|---|
| Pix2Pix | 0 | 282.35 | 0.383 ± 0.005 | 0.381 ± 0.052 | 0.190 ± 0.134 | 0.277 ± 0.052 | 8.46 ± 1.18 |
| RFC-GAN_SA7_EG3 | 0 | 396.35 | 0.583 ± 0.005 | 0.413 ± 0.058 | 0.268 ± 0.183 | 0.271 ± 0.034 | 7.78 ± 1.31 |
| RFC-GAN_SA12_EG3 | 0 | 250.92 | 0.331 ± 0.004 | 0.361 ± 0.052 | 0.171 ± 0.122 | 0.361 ± 0.064 | 8.95 ± 1.22 |
| RFC-GAN_SA12_EG3_R1 | 0 | 257.96 | 0.346 ± 0.004 | 0.368 ± 0.049 | 0.168 ± 0.134 | 0.344 ± 0.060 | 8.76 ± 1.15 |
| RFC-GAN_SA12_EG3_R1 | 5 | 218.60 | 0.277 ± 0.003 | 0.370 ± 0.076 | 0.222 ± 0.168 | 0.366 ± 0.070 | 8.82 ± 1.76 |
| RFC-GAN_SA12_EG3_R1 | 10 | 220.49 | 0.279 ± 0.003 | 0.376 ± 0.075 | 0.246 ± 0.177 | 0.380 ± 0.065 | 8.68 ± 1.75 |
| RFC-GAN_SA12_EG3_R1 | 15 | 221.59 | 0.285 ± 0.003 | 0.376 ± 0.072 | 0.253 ± 0.178 | 0.389 ± 0.060 | 8.67 ± 1.70 |
| RFC-GAN_SA12_EG3_R1 | 20 | 226.25 | 0.297 ± 0.003 | 0.377 ± 0.070 | 0.258 ± 0.180 | 0.390 ± 0.056 | 8.62 ± 1.66 |
| RFC-GAN_SA9_EG3_R1 | 0 | 411.95 | 0.605 ± 0.012 | 0.414 ± 0.062 | 0.216 ± 0.140 | 0.263 ± 0.083 | 7.76 ± 1.31 |
| RFC-GAN_SA9_EG3_R1 | 5 | 362.69 | 0.497 ± 0.013 | 0.399 ± 0.067 | 0.234 ± 0.174 | 0.293 ± 0.082 | 8.10 ± 1.50 |
| RFC-GAN_SA9_EG3_R1 | 10 | 274.08 | 0.363 ± 0.007 | 0.373 ± 0.066 | 0.228 ± 0.165 | 0.372 ± 0.061 | 8.69 ± 1.54 |
| RFC-GAN_SA9_EG3_R1 | 15 | 260.25 | 0.363 ± 0.004 | 0.364 ± 0.062 | 0.225 ± 0.163 | 0.404 ± 0.051 | 8.89 ± 1.46 |
| RFC-GAN_SA9_EG3_R1 | 20 | 263.26 | 0.374 ± 0.005 | 0.359 ± 0.058 | 0.223 ± 0.158 | 0.413 ± 0.049 | 9.00 ± 1.38 |
| RFC-GAN_SA7_EG3_R1 | 0 | 281.87 | 0.387 ± 0.006 | 0.373 ± 0.047 | 0.194 ± 0.137 | 0.299 ± 0.055 | 8.64 ± 1.10 |
| RFC-GAN_SA7_EG3_R1 | 5 | 277.86 | 0.367 ± 0.004 | 0.374 ± 0.065 | 0.216 ± 0.169 | 0.346 ± 0.064 | 8.67 ± 1.49 |
| RFC-GAN_SA7_EG3_R1 | 10 | 256.22 | 0.334 ± 0.004 | 0.376 ± 0.068 | 0.244 ± 0.178 | 0.383 ± 0.056 | 8.64 ± 1.57 |
| RFC-GAN_SA7_EG3_R1 | 15 | 268.10 | 0.362 ± 0.005 | 0.378 ± 0.064 | 0.256 ± 0.183 | 0.388 ± 0.051 | 8.58 ± 1.50 |
| RFC-GAN_SA7_EG3_R1 | 20 | 284.71 | 0.399 ± 0.006 | 0.382 ± 0.062 | 0.265 ± 0.185 | 0.385 ± 0.046 | 8.49 ± 1.44 |

Table 5: Ablations across self-attention (SA), edge-gating (EG3), and recursive conditioning (R1) combinations.

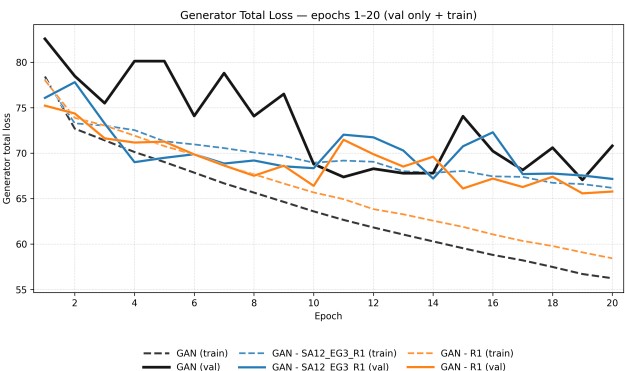

Figure 7: Validation loss curves comparing the SA12–EG3–R1 ensemble with the R1-only model, highlighting both as strong contenders against the baseline.

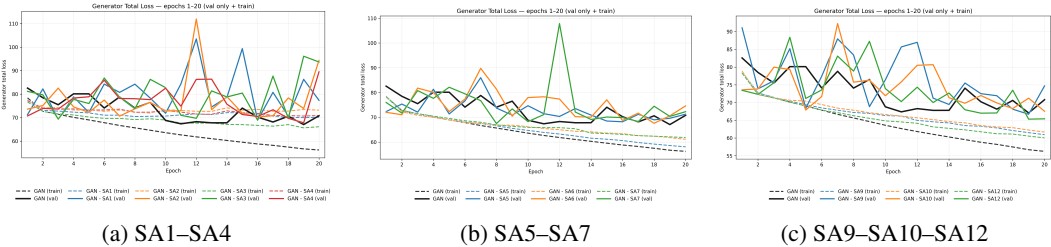

(a) SA1–SA4        (b) SA5–SA7        (c) SA9–SA10–SA12

Figure 8: Self-attention placement ablations, with (c) showing the best results and (a) showing the worst.

