# OpenReview forum: "RFC-GAN — Feedback-Conditioned GAN for Iterative Memory-to-Sketch Facial Translation"
_ICLR.cc/2026/Conference — ICLR 2026 Conference Desk Rejected Submission_

### Official Review · Reviewer_Mnia · 2025-10-23

**Soundness:** 2
**Presentation:** 2
**Contribution:** 1
**Rating:** 2
**Confidence:** 4

**Summary:**

The approach essentially follows the pix2pix framework. In my assessment, the authors primarily apply existing techniques to the sketch synthesis task for testing and analysis. Therefore, the work demonstrates **limited novelty and advancement**.

**Strengths:**

The approach essentially follows the pix2pix framework. In my assessment, the authors primarily apply existing techniques to the sketch synthesis task for testing and analysis.

**Weaknesses:**

### 1. Limited Novelty
- The introduction of EG (Edge Guidance) is presented as a fine-tuning strategy, but the main text lacks a concrete mathematical formulation or detailed workflow.
- The concept of EG itself is cited from Tang et al., further limiting the perceived novelty.

### 2. Method Generality and Specificity
- The proposed method does not demonstrate a clear, specific relationship to sketch synthesis.
- It is unclear how the method would perform if applied to other image translation tasks.

### 3. Literature Review
- The references consist mainly of older works.
- The survey of recent advancements in **GANs** and **sketch synthesis** is insufficient.

### 4. Paper Organization
- The structure and presentation of the paper are unusual and inconsistent with common practices, which hinders the understanding of its logical flow.

### 5. Technical Concerns regarding Self-Attention (SA)
- The paper uses SA only in later layers, while many efficient SA methods exist that could be applied in earlier layers. An analysis of using SA in earlier layers is necessary.
- The novelty of introducing SA is questionable, as:
    - SAGAN already introduced SA into GANs.
    - Many subsequent works, including diffusion models, have incorporated SA into U-net architectures.
- The significant increase in computational complexity due to SA raises concerns about its cost-effectiveness relative to any performance gains.

**Questions:**

### Core Novelty
1.  Please clearly state the core innovation of this work and its essential difference from the standard pix2pix framework.
2.  Can you provide a precise mathematical definition or algorithmic workflow for the EG (Edge Guidance) strategy? What are the specific improvements compared to the method by Tang et al.?

### Method Design
3.  Why is the proposed method particularly suitable for sketch synthesis? Has its generality been verified on other image translation tasks?
4.  Why is the self-attention mechanism only used in the deeper layers of the network? Were more efficient self-attention modules explored for the earlier layers?

### Experimental Analysis
5.  Does the performance gain from using self-attention justify its computational overhead? Please provide a detailed complexity vs. performance trade-off analysis.
6.  Can comparisons with more recent sketch generation or image translation methods be added to better demonstrate the advantages of the proposed approach?

### Suggestions
*   **Literature Review**: It is recommended to supplement the review with the latest advances in GANs and sketch synthesis.
*   **Paper Structure**: It is recommended to adjust the paper's structure to align with conventional academic logic for better readability.

---

### Official Review · Reviewer_x4GL · 2025-10-30

**Soundness:** 3
**Presentation:** 2
**Contribution:** 2
**Rating:** 2
**Confidence:** 4

**Summary:**

This paper presents RFC-GAN, a feedback-conditioned generative adversarial network designed to iteratively refine facial reconstructions from sketches or eyewitness recollections. The model extends the Pix2Pix framework by introducing a Recursive Feedback Conditioning (RFC) mechanism in which both the generator and discriminator incorporate the previous output as an auxiliary input, enabling iterative self-improvement across inference steps. Two complementary modules, Self-Attention (SA) to enhance global feature consistency and Edge-Gating (EG) to preserve contour fidelity, are examined individually and in combination. Experiments on the CelebAMask-HQ synthetic sketch dataset and the CUHK hand-drawn sketch dataset demonstrate consistent gains across standard perceptual and structural metrics including FID, KID, LPIPS, SSIM, and PSNR, with the RFC-only configuration yielding the most stable and accurate reconstructions. The study positions feedback conditioning as a viable strategy for human-like iterative refinement in generative facial translation with potential applications in forensic reconstruction.

**Strengths:**

* The feedback-conditioning mechanism is well-motivated as a human-analogous process of iterative self-correction.

* Decent ablations (RFC, SA, EG, and their combinations) are conducted, isolating the effect of each module.

* The RFC-only variant shows improvements across multiple perceptual metrics.

* The study addresses a socially meaningful application (forensic sketch translation), potentially encouraging follow-up research in this niche.

**Weaknesses:**

* The RFC design closely resembles known self-conditioning and recurrent refinement paradigms, offering limited conceptual advancement.

* Both datasets are small or synthetic. The model’s generalizability to unconstrained, hand-drawn sketches or diverse domains is untested.

* No comparisons are made against modern diffusion-based or transformer-based translators, which are now dominant in image synthesis tasks.

* The recursive conditioning lacks formal analysis of convergence or potential instability over multiple iterations.

*  The combination of SA and EG underperforms compared to the RFC-only model, suggesting redundancy and limited synergy.

* Occasional verbosity and inconsistent terminology reduce readability in sections describing architecture and results.

**Questions:**

1. How does RFC-GAN compare against diffusion-based iterative refiners or self-conditioning methods like SR3 or SelfRefine?


2. Did the authors test beyond the face-sketch domain (e.g., general image restoration or object sketches) to validate cross-domain adaptability?


3. Can the authors provide quantitative evidence of iterative convergence (e.g., change in FID per iteration) to support claims of refinement stability?


4. What are the computational implications of recursive conditioning? Does the feedback loop significantly increase training time or memory usage?


5. Could longer-term feedback (multiple previous outputs rather than one) be incorporated without instability?

---

### Official Review · Reviewer_hXX8 · 2025-11-01

**Soundness:** 3
**Presentation:** 2
**Contribution:** 1
**Rating:** 2
**Confidence:** 4

**Summary:**

The paper proposes RFC-GAN, a conditional image-to-image generative adversarial network that reconstructs photorealistic faces from eyewitness sketches through iterative refinement. The key idea is Recursive Feedback Conditioning, at every training and inference step the previous generator output is concatenated to the sketch and passed back into both generator and discriminator.

**Strengths:**

This paper proposes a feedback-conditioned generative framework and offers a principled approach to reliable memory-based facial reconstruction. The results show improvements in metrics such as FID.

**Weaknesses:**

1)Lack of motivation: Many diffusion-based methods, such as ControlNet[1], already perform well. The paper lacks discussion of these related works and does not clearly justify the significance of using Pix2Pix as the baseline for improvement.
2)Lack of comparison: Only vanilla Pix2Pix is evaluated. Stronger, task-specific baselines (e.g., diffusion or transformer models) are missing, making the practical advantage of RFC-GAN unclear.
3)Poor writing quality: The writing is disorganized, and the figures are unclear. It's really hard to follow.
[1] Zhang L, Rao A, Agrawala M. Adding conditional control to text-to-image diffusion models[C]//Proceedings of the IEEE/CVF international conference on computer vision. 2023: 3836-3847.

**Questions:**

Please refer to the weaknesses part.

---

### Official Review · Reviewer_fQWo · 2025-11-05

**Soundness:** 1
**Presentation:** 1
**Contribution:** 1
**Rating:** 2
**Confidence:** 5

**Summary:**

This work proposes a context-feedback training paradigm for image-to-image GANs. It is built on top of the Pix2Pix framework with Self-Attention and Edge-Gating modules. Experiments demonstrate improvements in FID and LPIPS.

**Strengths:**

It is an interesting topic that could have real life impact.

**Weaknesses:**

- It is unclear if this work is about memory-to-sketch or sketch-to-photo. It is expected to be a work for memory-to-sketch by looking at the title and the abstract. But it appears to be a sketch-to-photo work by looking at the adopted method (Sec. 4) and the evaluation metrics.
- It is unclear to me what feedback is used as a condition in the GAN model.
- It lacks an overview of the proposed method. It is not clear how the RFC, SA,  and EG modules are incorporated in the Pix2Pix model. The motivation and design details of these modules are missing as well.
- Figures (Figs 1 & 2). The texts are too small to be recognised.
-  It is hard to tell that the proposed method is effective by looking at the results.

**Questions:**

- see weaknesses.

---

### Note · Program_Chairs · 2026-01-17
**Submission Desk Rejected by Program Chairs**

The following references in this submission do not refer to real documents and/or have major errors in bibliographic information:

 Charlie D. Frowd. Facial composite systems: A review and survey of the effectiveness of different systems. Forensic Science International, 2012